# Drainage Practice of Rice Paddies as a Sustainable Agronomic Management for Mitigating the Emission of Two Carbon-Based Greenhouse Gases (CO$_2$ and CH$_4$): Field Pilot Study in South Korea

**Wonjae Hwang** [1] , **Minseok Park** [1] , **Kijong Cho** [2] **and Seunghun Hyun** [2,*]

1 Ojeong Resilience Institute, Korea University, Seoul 02841, Republic of Korea; hwj0145@korea.ac.kr (W.H.); asithinkyou@korea.ac.kr (M.P.)

2 Department of Environmental Science and Ecological Engineering, Korea University, Seoul 02841, Republic of Korea; kjcho@korea.ac.kr

* Correspondence: soilhyun@korea.ac.kr; Tel.: +82-2-3290-3068

**Abstract:** Rice is one of the staple foods in Asian countries, and rice paddies are significant sources of greenhouse gas (GHG) emissions in agricultural sectors. In addition, drainage practice has been recognized as a key factor influencing both rice production and GHG emissions. In this field pot study, the effect of drainage method (e.g., intermittent drainage (ID) and continuous flooding (CF)) on GHG (CO$_2$ and CH$_4$) emissions was determined from three Korean paddies (BG, MG, and JS series), varying soil properties such as soil texture, labile carbon, and mineral types. The emission of GHGs was evidently influenced by the drainage, depending on the paddy's redox (Eh) shift upon flooding events. The Eh decline upon flooding was slower in JS pot, where pore-water concentration of ferric and sulfate ions is the highest (~up to 3-fold) among three paddies. MG pot was 2- to 3-fold more percolative than the others and the Eh drop during the flooding period was the smallest (staying above −50 mV). In ID treatment, CH$_4$ emission (t CO$_2$-eq ha$^{-1}$ y$^{-1}$) was reduced in a wide range by 5.6 for JS pot, 2.08 for BG pot, and 0.29 for MG pot relative to CF, whereas CO$_2$ emissions (t CO$_2$-eq ha$^{-1}$ y$^{-1}$) were increased by 1.25 for JS pot, 1.07 for BG pot, and 0.48 for MG pot due to the enhanced oxidation of labile carbon. Grain yield and aboveground biomass production from ID were no less than those from CF ($p < 0.05$). Consequently, the increase in global warming potential (Σ GWP) by ID varied as the order of JS (37%) > BG (14%) > MG (~0%) pots, and the negligible effect observed for MG pot is due to the equivalent trade-off between CO$_2$ and CH$_4$. The different benefits of drainage practices among paddy pots is due to the redox response of paddy systems. The findings will be helpful to promote the efficacy of drainage practice on mitigating GHG emissions for the sustainable agronomic management of rice paddies in response to climate change.

**Keywords:** paddy soil; intermittent drainage; carbon dioxide; methane; sustainable rice cultivation; climate change

## 1. Introduction

The Intergovernmental Panel on Climatic Change (IPCC) reported that rice paddies are recognized as sources of considerable greenhouse gas (GHG) emissions. Rice (*Oryza sativa* L.) is one of the most important staple foods in Asian countries [1]. In particular, rice cultivation fields account for approximately 51% of the agricultural cropland area and 27% of the GHG emissions in Korean agricultural lands [2,3]. The common practice of rice cultivation involves maintaining a flooded environment by irrigation treatment from rice planting to harvest season [4,5]. To prepare for the crises caused by climate change, corresponding agricultural researches are needed to reduce GHG emissions while maintaining rice production. The flooding event comprises anoxic environments as the

biochemical activity reduces redox potential (Eh) of paddies [4]. A sufficiently low Eh of paddy water is required to invoke the formation of methane ($CH_4$), since methanogenic bacteria can only metabolize in strictly anoxic environments. Consequently, drainage treatment has been recognized to significantly affect $CH_4$ emissions from paddy soils [5]. The drainage has been widely recognized to suppress $CH_4$ emissions by increasing the Eh level of paddy fields [6–9]. For example, the implementation of intermittent drainage during the rice-growing season was found to reduce $CH_4$ emissions relative to continuous flooding in Korean paddy by about 48.5%.

Meanwhile, carbon dioxide ($CO_2$) emission from agricultural land has not been widely recognized as a GHG according to IPCC guidelines, because agricultural soils are generally regarded as a carbon sink [1]. However, $CO_2$ emissions from Korean paddy fields can be significant, because more than 50% of the aboveground rice biomass is not returned back to the field after harvesting, but is used for other industries (e.g., animal husbandry) [2].

Soil wetness is one of the most important environmental factors that controls the degree of soil microbial respiration; the irrigation practice in rice paddies can significantly affect soil basal respiration rates [10]. The general consensus from previous studies is that drainage increases atmospheric oxygen ($O_2$) diffusion into soils, thereby enhancing aerobic decomposition and promoting $CO_2$ production, while at the same time suppressing $CH_4$ emissions [5,11]. Thus, in order to precisely assess the benefit of intermittent drainage practice on GHGs emission, the trade-off between the reduced $CH_4$ emissions and the increased $CO_2$ emissions must be properly accounted for.

Paddy soils are subject to changes from oxic to anoxic conditions under flooding treatment, leading to decreasing soil redox (Eh) conditions and the sequential reduction of terminal electron acceptors, such as nitrate ($NO_3^-$), manganese ($Mn^{4+}$), ferric ($Fe^{3+}$), and sulfate ($SO_4^{2-}$), according to their energy release and availability [4]. For example, when $O_2$ is depleted and Eh decreases, the reduction of $NO_3^-$ to nitrogen gas occurs before $Mn^{4+}$ (in $MnO_2$) is reduced to $Mn^{2+}$. Then, $Fe^{3+}$ (in $Fe(OH)_3$) is transformed to $Fe^{2+}$ in the range of 200 mV < Eh < −100 mV, after which $SO_4^{2-}$ becomes the preferred terminal electron acceptor in highly reduced condition (e.g., −100 mV to −180 mV) [4,12]. If the Eh value drops as low as −180 mV, methanogenesis occurs and the remaining $CO_2$ is reduced with soil organic carbon (SOC) (e.g., acetic acid) to $CH_4$ by methanogens [4,5]. Therefore, the abundance of electron acceptors in paddy water that can be supplied from the soil minerals is known to slow down the rate of Eh reduction, resulting in suppressed $CH_4$ flux [13,14].

Percolation (or internal drainage) is the process by which water moves downward through the soil under gravitational forces [15]. The rate of percolation is controlled by soil particle sizes [16], and it measures how promptly paddy soil becomes unsaturated and the oxic condition during the drainage event. The effect of drainage practice on the migration of GHG emissions should markedly vary with the textural class of the paddy field. For example, Yagi et al. [17] observed an approximately 2.8-fold decrease in seasonal $CH_4$ fluxes from a loamy paddy field, as compared to a silty clay loam paddy field, due to a roughly 3-fold higher percolation rate of the former (2.5 cm $day^{-1}$ vs. 7.7 cm $day^{-1}$). The reduced $CH_4$ emission from the loamy field was explained by the occurrence of the prompt and wider Eh elevation upon draining paddy water. Therefore, $CH_4$ production could be suppressed in highly percolative paddy by repeated drainage events as the soil continuously formed in oxic conditions [5,15,17].

Global warming potential (GWP) was widely adopted to provide an understanding of agricultural impacts on radiative forcing [1]. This concept allows for direct comparisons of the overall impacts induced by GHGs. Another concept, greenhouse gas intensity (GHGI) which is calculated by dividing the GWP by the crop production, was also introduced to simultaneously assess GHG emissions and rice production in the paddy soils [18]. Existing literature shows that the emission of GHGs from paddy fields can be substantially reduced by adopting intermittent drainage practices [6–9], but no relationship between GWP and grain production has been delineated in combination with different soil physicochemical characteristics and irrigation methods. In addition, only a few studies have simultaneously

examined the emission dynamics of GHGs ($CO_2$ and $CH_4$) from rice paddies and their trade-offs by intermittent drainage [8,9].

So far, a good deal of laboratory and field studies reported that GHG emissions from rice paddy fields is greatly reduced by introducing intermittent drainage during conventional flooding practice. However, the effect of the intermittent drainage on the mitigation of GHG emissions from Korean rice paddies has not been well examined. Therefore, the primary goal of this study was to quantify the flux of GHGs (e.g., $CO_2$ and $CH_4$) as affected by irrigation method (e.g., continuous flooding vs. intermittent drainage) from three different Korean paddies over the duration of rice cultivation (from transplanting to harvest). Seasonal variation of GHGs flux was interpreted by analyzing the temporal condition of three paddy soils such as redox couple, redox potential, and soil wetness. By coupling rice grain yield and global warming potential, the benefit of the intermittent drainage practice was also addressed.

## 2. Materials and Methods

### 2.1. Experimental Settings

#### 2.1.1. Rice Pot Design

Paddy soils of three different rice fields (e.g., Bugog (BG), Meagog (MG), and Jisan (JS) series) were collected near Korea University Farm (KU farm hereafter) in Gyeonggi Province, Korea. These soils have been used for rice cultivation for a couple of decades. Collected samples were air-dried in KU farm. A pot experiment was set up in a paddy field (L = 35 m, W = 25 m) in KU farm (37°35′01″ N, 127°14′16″ E).

Eighteen holes with a depth of 30 cm were dug at a spacing 2 m × 3 m and the pallet and non-woven mat were laid on the bottom of each hole [19] in order to eliminate the influence of the underlying soil condition (moisture status, groundwater table, etc.) (Figure S1). A cylindrical rubber pot (68 cm diameter and 46 cm height) was placed in each hole and was filled with the soil; approximately 132 kg for BG pot, 131 kg for MG pot, and 141 kg for JS pot, close to their bulk density in the field (1.21 g cm$^{-3}$ for BG pot, 1.20 g cm$^{-3}$ for MG pot, and 1.29 g cm$^{-3}$ for JS pot). Six weepholes with 5 cm diameter were installed in the bottom of the cylindrical rubber pots (Figure S1) to facilitate drainage according to the percolation properties of each pot. The pot experiment was conducted from May to September in 2019. This region belongs to temperate monsoon climate zone with an annual mean temperature of 12.0 ± 0.4 °C and a mean annual precipitation of 1408 ± 256 mm over the past 20 years. Intensive rainfall occurs between June and August (Figure S2).

#### 2.1.2. Soil Characterization

Selected properties of the paddy soils were characterized as follows; pH at a soil mass to solution volume ratio of 1:5 (g:mL), particle size distribution using the pipette method, percolation rate using air-entry permeameter method, total organic carbon by $K_2Cr_2O_7$ oxidation, and labile carbon by $KMnO_4$ oxidation, and Fe-oxide content by oxalate extraction (pH 3) and dithionite–citrate–bicarbonate (DCB) extractions [20,21]. Mineralogical compositions of clay particles (<2 μm) were identified using X-ray diffraction (XRD) with Cu Ka radiation at 45 kV and 40 mA (D/max 2500 v/pc, Rigaku, Tokyo, Japan). All patterns were collected at a scanning speed of 1.2° min$^{-1}$ over the $2\theta$ range of 5–70°. The Highscore Plus data analysis program (Malvern Panalytical, Malvern, Worcestershire, UK) which includes the standards of the International Centre for Diffraction Data was used to identify the observed digital diffraction pattern [22].

#### 2.1.3. Farming Method

In a paddy pot, urea was applied as a N source (i.e., 70 kg ha$^{-1}$) and approximately 2 cm-depth of water was supplied before transplanting on 12 May. Four to five hills of 25-days-grown rice seedlings (*Oryza sativa* L., maturity 124 days) were manually transplanted on 16 May at a planting density of 0.15 m × 0.15 m. Paddy pots were initially flooded with 1–2 cm water until the rice growth attains the three-leaf stage on 30 May. To

conduct continuous flooding (CF) treatment, 5–7 cm depth of paddy water was maintained during irrigation period (31 May–10 August). For intermittent drainage (ID) method, the pots were intentionally drained three times (e.g., 31 May to 4 June, 16 June to 23 June, and 6 July and 28 July). The number of drainages was designed not to affect the rice yield based on our preliminary test. Water depth was measured daily during the irrigation period, and then deficit water was supplied to maintain the target water depth for each irrigation schedule, as shown in Table S1. For both irrigation methods, all paddy pots were permanently drained through the drain valve (Figure S1) on 10 August, and rice was harvested on 17 September which is 38 days after the final draining event. Agronomic practices conducted for rice pot experiment are summarized in Table S1. Three replicate experiments were performed in three paddy pots for each treatment.

### 2.2. Measurement and Data Collection

For 125 days after rice transplanting, the greenhouse gas, paddy water, and soil volume wetness data were collected on the same day at average intervals of 4 days. Details for the data collection and analysis method follow.

#### 2.2.1. Greenhouse Gas Emission

The flux of two GHGs ($CO_2$ and $CH_4$) was measured using the closed chamber method [19]. The system consists of a closed chamber with a measurement unit in which a moisture filter, direct current (DC) pump, flow meter, gas detector module, and data logger are sequentially connected (Figure S1). An opaque acrylic cylinder, which had a diameter of 30 cm and 40 cm in height (or 100 cm in height depending on rice growth), was anchored into the soil surface to collect gas emitted from the paddy system, including paddy soils and rice plants. The chamber was wrapped with an aluminum foil to minimize air temperature changes inside the chamber during the period of sampling. The chamber was able to cover the four rice plants transplanted in each pot without air exchange with the outside. A DC pump (Motorbank, Seoul, Republic of Korea) and an air flow meter (Dwyer, Michigan City, IN, USA) were installed to maintain a constant flow rate ($\cong 1$ L min$^{-1}$) of air between the chamber and the detector, thereby forming a continuous air circulation system. The concentration of $CO_2$ and $CH_4$ were directly determined using a $CO_2$ sensor (Soha-Tech, Seoul, Republic of Korea) and $CH_4$ sensor (Axetris, Kaegiswil, Switzerland) incorporated within the measuring device.

The measurement was conducted twice on a given day (e.g., between 8 and 11 a.m. and between 3 and 6 p.m.) and results were averaged for daily emissions based on sampling protocols [23]. The amount of gas emission (gas flux, *F*, mg m$^{-2}$ h$^{-1}$) was calculated using Equation (1) [24].

$$F = \rho \cdot \left[\frac{V}{A}\right] \cdot \left[\frac{\Delta C}{\Delta t}\right] \cdot \left[\frac{273}{(T+273)}\right] \qquad (1)$$

where $\rho$ is the density of gas (mg m$^{-3}$), *V* is the volume of the chamber (m$^3$), *A* is the bottom area of the chamber (m$^2$), $\Delta C / \Delta t$ is the average rate of concentration change (ppmV h$^{-1}$), and *T* is the average temperature in the chamber (°C).

#### 2.2.2. Chemical Analysis of Paddy Water and Paddy Soil

A paddy water sample was filtered through a 0.45 μm membrane filter prior to chemical analysis. The concentration of $Fe^{3+}$ and $Fe^{2+}$ was determined at 510 nm using a UV–Vis spectrophotometer (UV-1800, Shimadzu, Kyoto, Japan) using 1,10-phenanthroline [21]. The concentration of $SO_4^{2-}$ was determined by ion chromatography (ICS-2000, Dionex, Sunnyvale, CA, USA) with anion-exchange column (IonPac AS18, 250 mm × 4 mm, Dionex, Sunnyvale, CA, USA). Temporal change in two redox elements (Fe and S) was assumed to be due to the result of redox potential of paddy pots.

On each day when GHGs flux was measured, pH and Eh were measured for pore water at a 5 cm-depth from soil-water interface using a portable pH/Eh meter (Pro Plus, YSI, Yellow Springs, OH, USA). The volume wetness ($v/v$, %) of paddy soils at 5 cm and 15 cm depth was also measured using a soil moisture sensor (HydraProbe, Stevens, Portland, OR, USA).

### 2.2.3. Rice Grain and Biomass

Rice plant samples were taken from three sample areas of 0.36 m$^2$ in each plot where the grain yields were determined at maturity stage (17 September). At the same time, three hills of straw at each plot were randomly sampled and oven-dried at 85 °C to a constant weight to obtain the aboveground biomass. The rice grain was adjusted to the standard moisture content of 0.14 g H$_2$O g$^{-1}$ (fresh weight) to obtain the grain production [9]. The production of grain and aboveground biomass were expressed as the unit of t ha$^{-1}$ considering the pot surface area.

### *2.3. Data Analysis*
### 2.3.1. GWP, Cumulative GHGs Emissions, and GHGI Estimates

The GWP (kg CO$_2$-eq ha$^{-1}$) based on the CO$_2$ and CH$_4$ emissions were used to account for the climatic impact of among three soils under different irrigation methods. The GWP coefficient is 34 for CH$_4$ when the GWP value for CO$_2$ is taken as one [1].

The cumulative emissions of CH$_4$ and CO$_2$ (t CO$_2$-eq t grain$^{-1}$) were calculated for each plot according to the following equation, as described by Equation (2) [9].

$$\text{Cumulative emission} = \Sigma[(F_i + F_{i+1})/2 \times d] \tag{2}$$

where $F_i$ and $F_{i+1}$ are the measured fluxes of two consecutive sampling days (t CO$_2$-eq ha$^{-1}$), and $d$ is the number of days between two sampling days.

The term GHGI (t CO$_2$-eq ha$^{-1}$) relates the sum of GWP production to grain yield (t ha$^{-1}$) for each paddy pot and was calculated by Equation (3) [18].

$$\text{GHGI} = \text{GWP}/\text{grain production} \tag{3}$$

### 2.3.2. Statistical Analysis

Statistical differences (*t*-test) were determined at the significance level of $p < 0.05$ between the CF and ID methods using SAS 9.4 (SAS Institute Inc., Cary, NC, USA).

## 3. Results and Discussion
### *3.1. Soil Characteristics*

Selected properties of paddy soils used in this study are presented in Table 1. All soils exhibited near natural pH (6.6–7.0). The labile carbon content (g kg$^{-1}$) of three soils varied as follows; JS (10.6 ± 1.27) > BG (8.42 ± 0.44) > MG (5.48 ± 0.55). Because of the high sand content, MG pot showed the highest percolation rate (cm d$^{-1}$) as 8.51 ± 1.06, more than twice of others (BG pot = 2.84 ± 0.53 and JS pot = 3.90 ± 0.55). The soluble concentration of Fe$^{3+}$ (mg L$^{-1}$) had a similar range (0.11–0.16) across the three pots. JS pot showed over 2.6-fold higher SO$_4$$^{2-}$ concentration (66.5 ± 2.7 mg L$^{-1}$) than BG and MG pots (25.6 ± 0.3 and 22.7 ± 0.6 mg L$^{-1}$, respectively). The results of XRD analysis of clay fraction showed that JS soil is mainly composed of quartz (SiO$_2$), ferrihydrite (Fe(OH)$_3$), and langite (Cu$_4$(SO$_4$)(OH)$_6$·2H$_2$O), while both BG and MG mainly contain quartz (SiO$_2$) and magnetite (Fe$_3$O$_4$). Note that ferrihydrite is amorphous whereas magnetite is crystalline [25]. Likewise, the content of Ox–Fe was higher in JS soil (12.8 ± 0.32 g kg$^{-1}$) in comparison with BG and MG soils.

**Table 1.** Physical and chemical properties of three Korean paddy soils.

| Properties | Soils [a] | | |
|---|---|---|---|
| | BG | MG | JS |
| Order/Subgroup [b] | Ultisols/Typic Fragiudults | Inceptisols/Anthraquic Eutrudepts | Alfisols/Typic Endoaqualfs |
| pH [c] | $6.6 \pm 0.1$ | $6.8 \pm 0.1$ | $7.0 \pm 0.1$ |
| Sand [d] | $27.4 \pm 2.1$ | $54.5 \pm 0.3$ | $41.4 \pm 0.5$ |
| Clay [d] | $33.9 \pm 0.4$ | $18.6 \pm 1.0$ | $24.8 \pm 0.3$ |
| Percolation rate [e] | $2.84 \pm 0.53$ | $8.51 \pm 1.06$ | $3.90 \pm 0.55$ |
| TOC [f] | $14.4 \pm 0.40$ | $14.9 \pm 0.39$ | $21.2 \pm 0.46$ |
| Labile C [g] | $8.42 \pm 0.44$ | $5.48 \pm 0.55$ | $10.6 \pm 1.27$ |
| Soluble ferric [h] | $0.16 \pm 0.03$ | $0.11 \pm 0.03$ | $0.15 \pm 0.02$ |
| Soluble sulfate [i] | $25.6 \pm 0.3$ | $22.7 \pm 0.6$ | $66.5 \pm 2.7$ |
| DCB-Fe [j] | $38.7 \pm 0.33$ | $32.3 \pm 0.73$ | $29.3 \pm 0.22$ |
| Ox-Fe [k] | $6.62 \pm 0.26$ | $5.84 \pm 0.05$ | $12.8 \pm 0.32$ |
| Major Minerals | Quartz ($SiO_2$), Magnesioferrite ($MgFe_2O_4$), Magnetite ($Fe_2O_4$) | Quartz ($SiO_2$), Gupeiite ($SiFe_3$), Magnetite ($Fe_2O_4$) | Quartz ($SiO_2$), Ferrihydrite ($Fe_2O_3 \cdot 0.5H_2O$), Langite ($Cu_4(SO_4)(OH)_6 \cdot 2H_2O$) |

[a] Three paddy soils were named by their respective series. [b] Soil classification of order and subgroup by soil taxonomy. [c] Aqueous soil pH measured at 1:5. [d] Percent (%) of sand and clay with particle sizes less than 2 mm and 2 μm, respectively. [e] Percolation rate (cm $day^{-1}$). [f] Total organic carbon (g $kg^{-1}$). [g] Labile carbon (g $kg^{-1}$). [h] Water soluble ferric ($Fe^{3+}$) measured at 1:5 (mg $L^{-1}$). [i] Water soluble sulfate ($SO_4^{2-}$) measured at 1:5 (mg $L^{-1}$). [j] Dithionite–citrate–bicarbonate extractable iron (g $kg^{-1}$). [k] Oxalate (pH 3) extractable iron (g $kg^{-1}$).

### 3.2. Weather Conditions of Experimental Site

Data of precipitation and air temperature during pot experiment (May–September) are provided in Figure S2. The mean temperature over the given period was $23.6 \pm 2.1$ °C, with the highest temperature between July and August. The cumulated precipitation was $731 \pm 45$ mm, and most of the rainfall occurred between August and September (Figure S2). Typical temperate monsoon climate was observed during the experimental period.

### 3.3. Impact of Irrigation Methods on Paddy Water

The concentration (mg $L^{-1}$) of $Fe^{2+}$ and $SO_4^{2-}$, and Eh (mV) for the paddy water of three paddy pots (BG, MG, and JS) under two irrigation methods (CF and ID) are displayed as a function of time in Figure 1. Note that the CF pot is continuously submerged during the irrigation period (12 May–10 August) while the ID pots are intentionally drained and re-flooded three times during the given period. Both CF and ID pots were permanently drained after 10 August. The gray area in Figure 1 indicates the duration of (re)flooding in the ID method.

In the CF pot, the concentration of $Fe^{2+}$ gradually increased and remained relatively constant over the flooded period (12 May–10 August) with an average of $0.27 \pm 0.02$, $0.14 \pm 0.02$, and $0.44 \pm 0.05$ mg $L^{-1}$ for BG, MG, and JS pots, respectively (Figure 1a, Figure 1b, and Figure 1c). The increase in $Fe^{2+}$ can be assumed to be a result of reduction of $Fe^{3+}$ ion in flooded paddies [26]. The highest $Fe^{2+}$ concentration in the JS pot can be explained by the different mineralogical compositions of the three paddy soils [12]. Fe-oxides are the most common source of iron in soils [27]; thus, the Fe level in paddy water varies with the type of Fe oxides present in paddy soils [28]. As shown in Table 1, ferrihydrite is a major Fe-oxide in JS soil and Ox-Fe content is almost two-times greater than the others, whereas magnetite is dominant in both BG and MG soils. Note that short-range ordered Fe-oxide (i.e., ferrihydrite) has lower thermodynamic stability than long-range ordered one (i.e., magnetite). For example, the solubility of $Fe^{3+}$ maintained by ferrihydrite is 89 times greater than that maintained by magnetite [25] and the reduction potential of ferrihydrite (Eh = 12 mV at pH 7) is considerably higher than that of magnetite (Eh = $-310$ mV at pH 7) [27].

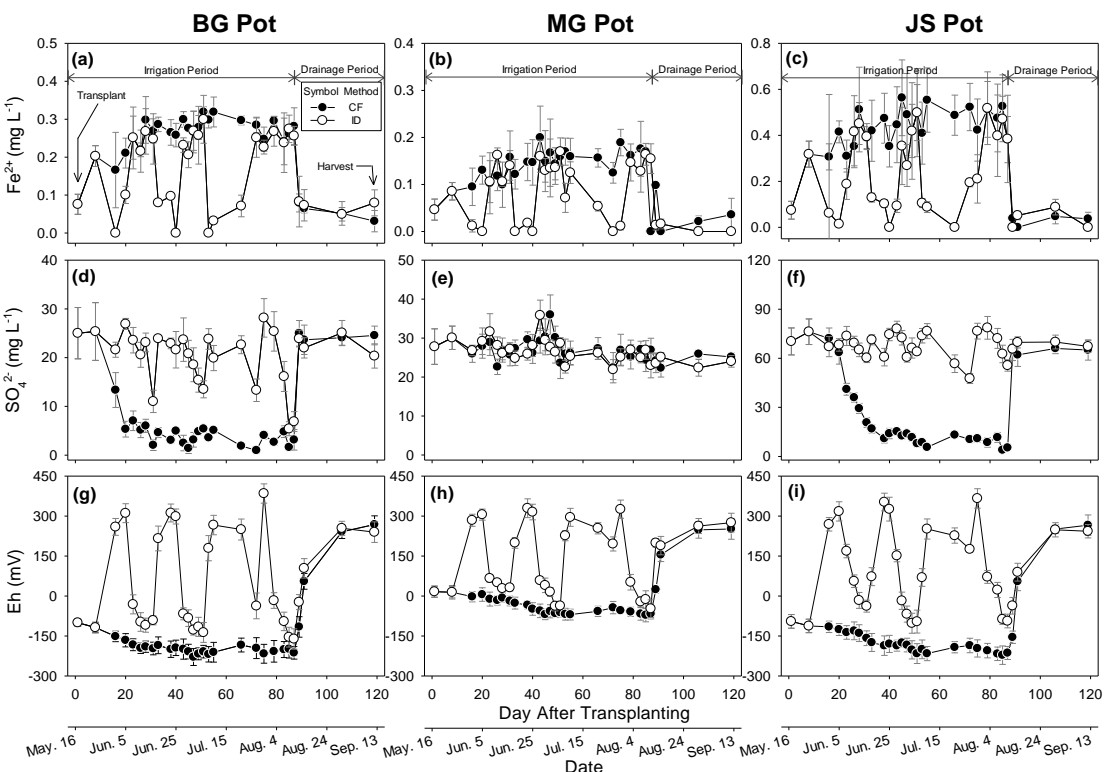

**Figure 1.** Concentration (mg L$^{-1}$) of Fe$^{2+}$ (**a–c**) and SO$_4{}^{2-}$ (**d–f**) and the magnitude (mV) of Eh (**g–i**) in paddy water of BG, MG, and JS pots under the method of continuous flooding (CF) and intermittent drainage (ID). White and gray columns represent drained and flooded periods of ID method, respectively. All pots were permanently drained on 10 August. The duration of the irrigation (12 May–10 August) and the timings of transplant (16 May) and harvest (17 September) are also shown.

Meanwhile, the concentration (mg L$^{-1}$) of SO$_4{}^{2-}$ in paddy water tended to decrease and the declining pattern was different among pots (Figure 1d–f). The initial SO$_4{}^{2-}$ concentrations were 25.0 $\pm$ 5.3, 27.8 $\pm$ 4.6, and 70.3 $\pm$ 8.2 mg L$^{-1}$ in BG, MG, and JS pots, respectively. The concentration dropped to near 10 mg L$^{-1}$ after 20 days and 40 days for BG and JS pots, respectively. The Eh of respective pots also concurrently dropped to below −180 mV (Figure 1g,i) by that time, supporting the conjecture that the depletion of SO$_4{}^{2-}$ in paddy water is due to the reduction of SO$_4{}^{2-}$ under the reducing condition. In general, the time required for paddy water to attain an extremely reduced condition (e.g., <180 mV; CO$_2$ reducing) depends on the availability SO$_4{}^{2-}$ and the duration of aqueous SO$_4{}^{2-}$ depletion in paddy water and the activity of SO$_4{}^{2-}$-reducing bacteria becomes predominated in the Eh range between −100 mV and −180 mV [12]. In this study, the JS pot took approximately 20 days longer to attain the extremely reduced condition (e.g., −180 mV) than the BG pot because of the initial 2.6-fold higher SO$_4{}^{2-}$ concentration in the JS pot than the BG pot. On the other hand, the depletion of SO$_4{}^{2-}$ was not apparent in the MG pot (Figure 1e) as the Eh was persistently kept above −72 mV during the same period (Figure 1h). Since the MG pot is highly percolative (Table 1), frequent refurnishing of fresh water was needed to sustain—the 5–7 cm of paddy water level. Consequently, a weakly reduced condition was likely achieved in the MG pot (−72 mV < Eh < 16 mV) through continuous O$_2$ influx by fresh irrigation water supply.

In ID pots, on the contrary, the concentrations of Fe$^{2+}$ and SO$_4{}^{2-}$ and Eh values fluctuated in response to the occurrence of drainage and re-flooding events (Figure 1). For example, the concentration of Fe$^{2+}$ promptly decreased (below 0.1 mg L$^{-1}$) whereas the concentration of SO$_4{}^{2-}$ increased up to initial level by performing drainage. Simultaneously, the Eh rose up to 300 mV, indicating the development of oxic paddy condition in drained

paddies. The Eh dropped quickly during the re-flooding periods but persisted above the Eh of CF pots.

To explain thermodynamic stability of Fe-oxides and $SO_4^{2-}$ in paddy water, pH-Eh data collected from ID pots was plotted in the Pourbaix diagrams of Fe (solid line) and S (dashed line) species in Figure S3 [29]. Under pH-Eh range of this study, crystalline Fe-oxides present in BG and MG paddies (e.g., hematite and magnetite) are highly stable (Figure S3a) while in JS paddy the dissimilatory iron release from amorphous Fe-oxide (e.g., ferrihydrite) is thermodynamically favorable (Figure S3b). The pH-Eh sets of re-flooded paddies lie close to the border where $SO_4^{2-}$ convert to its reduced forms (e.g., $H_2S$ and $HS^-$). According to this theoretical reason, therefore, it can be reasonably assumed that (1) the highest concentration of $Fe^{2+}$ in JS paddy is due to the reduction of iron released from amorphous Fe-oxide present in the JS soil, and (2) the depletion of $SO_4^{2-}$ in flooded water is most likely due to the reduction of sulfur in reduced condition. The presence of these electron acceptors is known to slow the rate of Eh decrease in flooded paddies [13,14].

### 3.4. Impact of Irrigation Methods on GHGs Emission

Temporal change in GHG flux (kg $CO_2$-eq $ha^{-1}$ $day^{-1}$) measured for three paddy pots (BG, MG, and JS) under two irrigation methods (CF and ID) are present in Figure 2. Soil volumetric soil wetness (*v/v*, %) determined at the depth of 5 and 15 cm is shown in Figure S4. All CF pots were retained near saturated soil wetness (~40%) at 5 cm depth over the irrigation period. The soil wetness of ID pots gradually decreased as low as 15–20% and the wetness at 5 cm depth was always lower than that at 15 cm during drainage periods. The wetness difference between two depths was minimal for the MG pot due to its high percolation rate. Notice that the occurrence of Eh rise (Figure 1) is coincident to the timing of soil wetness decreases (Figure S4).

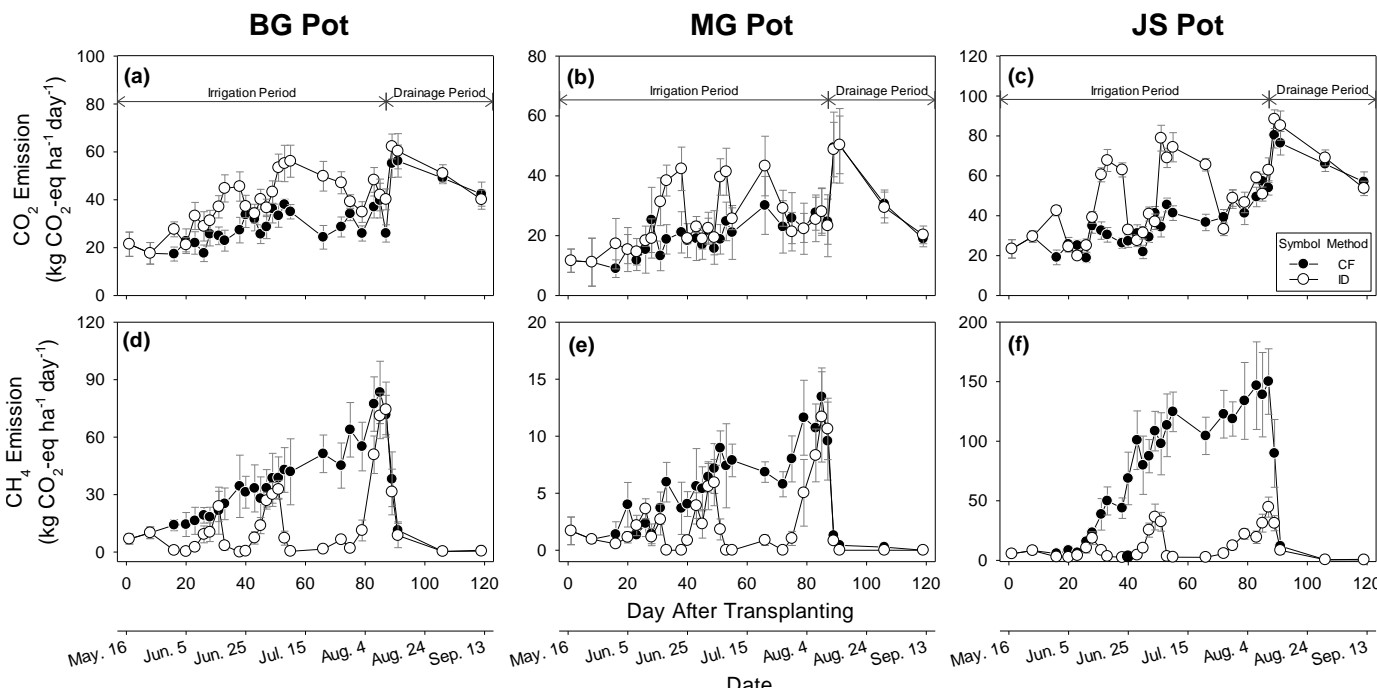

**Figure 2.** Emission (kg $CO_2$-eq $ha^{-1}$ $day^{-1}$) of $CO_2$ (**a**–**c**) and $CH_4$ (**d**–**f**) measured from BG, MG, and JS pots under the method of continuous flooding (CF) and intermittent drainage (ID). White and gray columns represent drained and flooded periods of ID method, respectively. All pots were permanently drained on 10 August.

### 3.4.1. Carbon Dioxide Emissions

The $CO_2$ flux was initially low until the three-leaf stage on 30 May and then peaked at the end of the irrigation period on 10 August, after which it gradually declined until rice maturity (Figure 2a–c). The pattern of $CO_2$ release from agronomic soils is correlated with the life cycle of crop growth, likely because mature plants have more root exudates for soil microbes [30].

The two irrigation treatments showed distinctively different $CO_2$ dynamics upon starting flooding practice at the three-leaf stage on 30 May. Over the cropping season, the $CO_2$ flux (kg $CO_2$-eq $ha^{-1}$ $day^{-1}$) under the CF method ranged from 17.3 to 39.7, 9.0 to 30.1, and 18.8 to 57.2 in BG, MG, and JS pots, respectively. The flux remained relatively low and then increased after permanent drainage starts, approximately five weeks before rice harvest. The $CO_2$ flux under the ID method ranged from 17.6 to 56.1, 11.2 to 43.3, and 19.8 to 78.9 in BG, MG, and JS pots, respectively, which present wider fluctuations than the CF method as affected by the repetition of draining and re-flooding situation. The $CO_2$ emissions appeared largest in JS paddy in which the content of labile carbon is highest among the three paddies (Table 1) [10]. Compared between two irrigation methods, the $CO_2$ flux of drained paddy in the ID method overwhelmingly exceeded that of flooded paddy in the CF method (Figure 2). It is well understood that the rhizosphere system will be exposed to sufficient $O_2$ upon draining paddy water, which in turn facilitates the rhizospheric respiration and aerobic microbial SOC degradation; thus, leading to significant augmentation of $CO_2$ production [14,31], compared to flooded conditions.

### 3.4.2. Methane Emissions

In the CF method, the $CH_4$ flux gradually increased steadily with rice growth until the end of flooding period after which it declined rapidly (Figure 2d–f). The $CH_4$ flux (kg $CO_2$-eq $ha^{-1}$ $day^{-1}$) in CF methods peaked as high as $83.3 \pm 16.3$ for the BG pot, $13.5 \pm 2.5$ for the MG pot, and $150.1 \pm 27.5$ for the JS pot in mid-August, after rice heading. In the ID method, the $CH_4$ flux substantially declined in drained paddies as 2.5–74.3, 1.2–11.7, and 4.1–44.6 in BG, MG, and JS pots, respectively. This observation agrees well with many previous results that intermittent drainage during the rice growing season can reduce $CH_4$ emissions from rice paddy fields by converting the redox potential to a more oxic condition [6,9,32].

Meanwhile, it is also worth noting that the $CH_4$ emission in the JS pot was substantially lower in three re-flooded periods (4 June–15 June, 24 June–5 July, and 29 July–10 August in Figure 2) compared to the given periods of the CF method. During these periods, both the average and peak of $CH_4$ flux were greatly decreased in the JS pot. Such a decrease was not pronounced in the other two pots. The contrast between pots can be explained by the different availability of electron acceptors ($Fe^{3+}$ and $SO_4^{2-}$), which regulates the pace of redox processes in rice paddies. Notice for the JS pot that the rise of Eh in the drained condition is the highest, whereas the drop of Eh is slower in the re-flooded condition (Figure 1). As previously mentioned for the JS pot, the dissimilatory Fe release from amorphous Fe-oxides is more likely and the concentration of $SO_4^{2-}$ is persistently highest among three paddy pots (Figure 1f). Consequently, the abundance of these electron acceptors essentially was able to slow the abrupt decrease in Eh in re-flooded JS paddy [12,13], thereby suppressing $CH_4$ production. For these three periods, log of $CH_4$ flux was well correlated with the Eh of paddy water (Figure 3). The variation of Eh in JS paddy was largest, ranging between $CO_2$-reducing conditions and $Fe^{3+}$ reducing conditions, whereas the Eh of MG paddy was ranging only in $Fe^{3+}$ reducing conditions. This indicates that impact of drainage on the Eh rise is pronounced in the JS pot, but negligible in the MG pot; so is the $CH_4$ mitigation. The Eh of the BG pot fell below the $CO_2$-reducing condition when flooded, and it increased in the range of <0 mV when drained.

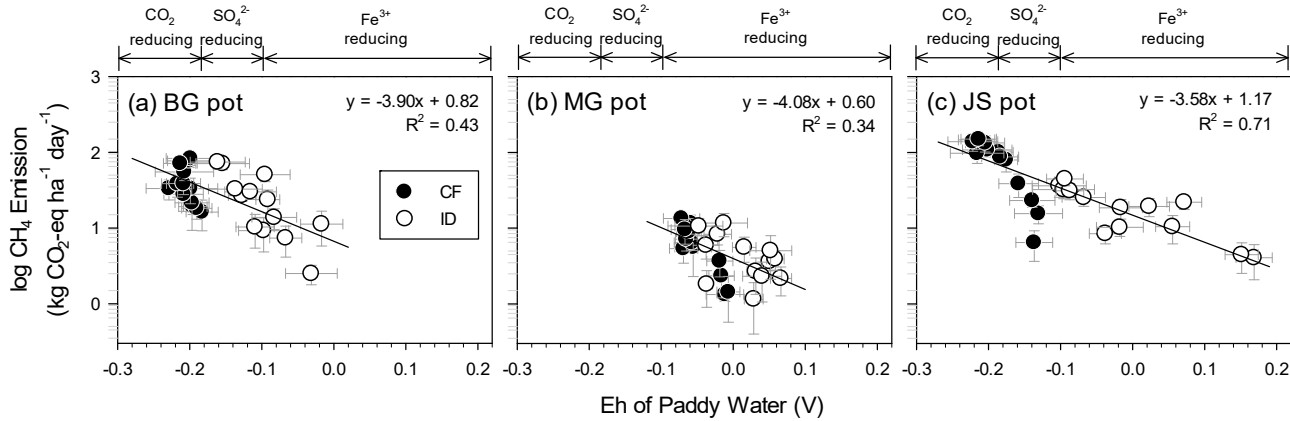

**Figure 3.** Correlation between the Eh in paddy water (V) and the log $CH_4$ emission (kg $CO_2$-eq ha$^{-1}$ day$^{-1}$) from flooded paddies of both continuous flooding (CF) and intermittent drainage (ID) method. Data point is the mean value of triplicates with the standard deviations shown as error bars. The solid line is regression fit and $R^2$ is the coefficient of determination.

### 3.5. Rice Yield

The crop growth (rice grain yield and aboveground biomass) determined for all paddy pots are presented in Table 2. The production of both grain and aboveground biomass in CF pots were in the following order: JS > BG > MG. The rice grain yield harvested in ID pots was slightly lower than CF pots, but the difference was not statistically significant. The growth data indicates that the drainage loss of the essential elements (e.g., N and C) in ID paddies is insufficient to inhibit the rice growth [33]. Similarly, Liu et al. [7] showed with meta-analysis data that the intermittent drainage has no effect on rice yield as long as the SOC level and N content (e.g., 50 kg N ha$^{-1}$) are sufficiently high, due to the enhanced decomposition of SOC in drained paddy. In general, the differences in crop production can be attributed to P availability in soils [4,34]. In this study, the concentration of available P determined for CF paddies was higher than ID paddies, primarily due to the enhanced reductive dissolution of P-bearing minerals in flooded paddies [4,34]. However, the level of P in these paddies satisfies the required level (e.g., 13.5 mg kg$^{-1}$) for rice paddies recommend by Korean RDA. Even though it is apparent that the availability of nutrients (e.g., N and P) in paddies is influenced by drainage practice, its impact on rice production was not pronounced under the experimental settings in this study.

**Table 2.** Data [a] of greenhouse gases (GHGs) emissions, rice production, and greenhouse gas intensity (GHGI) determined from the three different soils under two irrigation methods.

| Paddy Pot | Irrigation Method [b] | GHGs Emission (t $CO_2$-eq ha$^{-1}$) | | | Rice Production (t ha$^{-1}$) | | GHGI [d] (t $CO_2$-eq t grain$^{-1}$) |
|---|---|---|---|---|---|---|---|
| | | $CO_2$ | $CH_4$ | Σ GWP [c] | Grain | Aboveground Biomass | |
| BG | CF | 3.87 ± 0.12 | 3.24 ± 0.22 | 7.11 ± 0.25 | 7.09 ± 1.08 | 15.35 ± 2.06 | 1.00 ± 0.19 |
| | ID | 4.94 ± 0.15 (+28%) | 1.16 ± 0.09 (−64%) | 6.10 ± 0.17 (−14%) | 6.23 ± 0.96 (NS) [e] | 13.42 ± 2.35 (NS) | 0.98 ± 0.18 (NS) |
| MG | CF | 2.77 ± 0.21 | 0.48 ± 0.03 | 3.26 ± 0.21 | 5.82 ± 0.86 | 12.68 ± 1.78 | 0.56 ± 0.12 |
| | ID | 3.25 ± 0.19 (+17%) | 0.19 ± 0.02 (−61%) | 3.44 ± 0.19 (NS) | 5.10 ± 0.79 (NS) | 12.35 ± 1.87 (NS) | 0.67 ± 0.14 (NS) |
| JS | CF | 5.05 ± 0.12 | 6.62 ± 0.33 | 11.67 ± 0.35 | 7.79 ± 0.55 | 16.04 ± 1.55 | 1.50 ± 0.15 |
| | ID | 6.30 ± 0.13 (+25%) | 1.02 ± 0.06 (−85%) | 7.32 ± 0.14 (−37%) | 6.98 ± 1.19 (NS) | 14.68 ± 2.17 (NS) | 1.05 ± 0.20 (−30%) |

[a] An average value with standard deviation after ± sign. [b] CF and ID denote continuous flooding and intermittent drainage, respectively. [c] Global warming potential (GWP) is the sum of $CO_2$ and $CH_4$ emissions as unit of $CO_2$-equivaents (kg $CO_2$-eq ha$^{-1}$). [d] The ratio between GHGs ($CO_2$ + $CH_4$) emission to rice grain production (kg $CO_2$-equivalents kg grain$^{-1}$). [e] Value in parentheses denotes the percentage reduction (−) or percentage increase (+) in rice production, GHGs emissions, and GHGI relative to the CF method. NS indicates not significant according to the *t*-test at the *p* < 0.05 level.

*3.6. Efficiency of Intermittent Drainage on GHGs Reduction*

Rice production (t ha$^{-1}$), the cumulative GHGs emission (t CO$_2$-eq ha$^{-1}$), and the associated index of GWP (t CO$_2$-eq ha$^{-1}$) and GHGI (t CO$_2$-eq t grain$^{-1}$) are presented in Table 2. In the CF method, the CO$_2$ and CH$_4$ emissions over the rice growing season were measured as 3.87 ± 0.12 and 3.24 ± 0.22 for the BG pot, 2.77 ± 0.21 and 0.48 ± 0.03 for the MG pot, and 5.05 ± 0.12 and 6.62 ± 0.33 for the JS pot. The magnitude of two GHGs production (e.g., JS > BG > MG) was consistent with the order of labile carbon content in paddy pots (Table 1). Note that soil organic carbon is the primary source of electrons in soils [35] and the liable fraction of soil organic carbon is likely to act as preferential electron donors for soil microorganisms involved in GHG-forming reactions [10].

In the ID method, the cumulative CH$_4$ emissions were greatly reduced as 64%, 61% and 85% of the CF method in BG, MG and JS pots, respectively, mainly due to lower CH$_4$ production from drained oxic paddies (Figure 2d–f). Moreover, the CH$_4$ emission was also suppressed in re-flooding paddies of the ID method as shown in Figure 2. In contrast, the cumulative CO$_2$ emissions increased by 28% and 25% in BG and JS pots, respectively, but the increment was insignificant in the MG pot. Consequently, the reduction (%) of Σ GWP was significant only in the JS pot. For both BG and MG pots, the decrease in CH$_4$ emission was quantitatively offset by the increase in CO$_2$ emission. From the literature review, Sapkota et al. [36] reported that a 14.3~81.2% decrement of CH$_4$ emission and 3.7~47.1% of the increment of CO$_2$ emissions can be achieved by adopting different irrigation management practices [36]. They concluded that optimizing irrigation may help to reduce CH$_4$ emissions and net GWP.

In terms of GWP budget, the above result suggests that the drainage practice is not an effective strategy for BG and MG pots. The BG pot showed a rapid transition to anoxic conditions ($-161$ mV < Eh < $-17$ mV) compared to other pots. In this case, it is recommended to apply amending materials with electron acceptors to slow the abrupt drop of Eh value during re-flooding period [13], suppressing CH$_4$ emission when the paddy is re-flooded. For the MG pot, the increase in CO$_2$ emission is considerably exceeded the decrease in CH$_4$ emission and Σ GWP was slightly greater in the ID method. In this case, intermittent drainage may not be recommended for a strategy for mitigating GWP.

The greenhouse gas intensity (GHGI) was calculated by diving Σ GWP by rice yield to estimate the magnitude of GHG emissions (t CO$_2$-eq ha$^{-1}$) per rice yield (t grain ha$^{-1}$) of a given rice cultivation (Table 2). The GHGI (t CO$_2$-eq t grain$^{-1}$) of three pots was in the order of JS (1.50 ± 0.15) > BG (1.00 ± 0.19) > MG (0.56 ± 0.12) in the CF method and was JS (1.05 ± 0.20) > BG (0.98 ± 0.18) > MG (0.67 ± 0.14) in the ID method. Consequently, the JS pot exhibited the 30% reduction of GHGI indicating that an introduction of ID practice can reduce 30% of GWP without changing rice production. For both BG and MG pots, the difference in GHGI between two irrigation methods was not significant because the difference of both grain production and Σ GWP was not statistically different (*p* = 0.05). Meanwhile, the emission of N$_2$O gas was not considered in this study because we focused on the trade-off between CO$_2$ and CH$_4$ emissions by two irrigation methods.

**4. Conclusions**

The results of this field pot study clearly demonstrate that the drainage practice can reduce the magnitude of GHGs emission, but the efficiency varied depending on the characteristics of paddy soils. The reduced CH$_4$ emission (%) by introducing drainage was enhanced by the slow transition of the Eh to anoxic range in re-flooded paddy. The benefit of CH$_4$ mitigation was largely counterbalanced by the increased amount of CO$_2$ emission from drained paddy. Therefore, it can be concluded that the net benefit by drainage practice is controlled by the balance of two GHGs emissions from a given paddy. The mitigation of GHG emissions by intermittent drainage practice was apparent. However, when rice grain yield was combined together, the profit of drainage practice was significant only when CH$_4$ reduction by drainage is significant. Therefore, the intermittent drainage practice should not be recommended for a paddy with low CH$_4$ emissions. The strategies for

applying intermittent drainage practice should be precisely deployed by collecting local paddy characteristics to maintain a sustainable agricultural system by lowering the GHG emissions while maintaining rice production.

**Supplementary Materials:** The following supporting information can be downloaded at: https: //www.mdpi.com/article/10.3390/su16072802/s1, Table S1: Method and timing of a farming practices in the experiment field under continuous flooding and intermittent drainage; Figure S1: The components of the $CO_2$ and $CH_4$ flux measuring system (not to scale); Figure S2: Changes in air temperature and precipitations during the rice growing season; Figure S3: Pourbaix diagrams of Fe and S (25 °C) system. The pH and Eh values measured in BG and MG pots containing magnetite are plotted in (a), and the pH and Eh values measured in JS pot containing ferrihydrite are plotted in (b) under the condition of intermittent drainage; Figure S4: Changes in soil wetness ($v/v$, %) measured at the depth of 5 cm from the soil surface. For the treatment of intermittent drainage, soil wetness at the 15 cm depth was also determined.

**Author Contributions:** Conceptualization, W.H. and S.H.; methodology, M.P. and W.H.; validation, W.H.; formal analysis, W.H.; resources, S.H. and K.C.; data curation, W.H.; writing—original draft preparation, W.H.; writing—review and editing, S.H.; supervision, S.H.; project administration, K.C.; funding acquisition, K.C. All authors have read and agreed to the published version of the manuscript.

**Funding:** Korea Environment Industry & Technology Institute (KEITI) through Climate Change R&D Pro-ject for New Climate Regime (RS-2022-KE002294), Core Research Institute Basic Science Research Program through the National Research Foundation of Korea (NRF) funded by the Ministry of Education (NRF-2021R1A6A1A10045235).

**Institutional Review Board Statement:** Not applicable.

**Informed Consent Statement:** Not applicable.

**Data Availability Statement:** Data are contained within the article.

**Acknowledgments:** This work was supported by Korea University Grant.

**Conflicts of Interest:** The authors declare no conflict of interest.

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
