# Peer review of "Drainage Practice of Rice Paddies as a Sustainable Agronomic Management for Mitigating the Emission of Two Carbon-Based Greenhouse Gases (CO2 and CH4): Field Pilot Study in South Korea"

_sustainability, doi:10.3390/su16072802_

Round 1

Reviewer 1 Report

Comments and Suggestions for Authors

This study presented a comparasion on effects of different irrigation practise on CH4 and CO2 emitted from 3 paddy fields by a pot experiment in 2019. overoall, I recommend a major revision, because some confusing errors existed, detailed comments are as follows:

abstract:

line 11, the research backgrounded need to be clear and showed in the first sentence of abstract.

line 14: to a different extent, this expression is not suitable in a scientific paper. how much is the extent, you need to avoid words like this.

keywords:

greenhouse gas should be deleted as CH4 and CO2 were also included in the keywords

introduction:

this part confused me somehow, 

Line 52 soil wetness is not usually seen, i think you should call it soil mositure instead. this should be revised all over the manuscript.

Line53: actually after i have read the whole paper, I found that when measuring the CO2, rice plants is also covered in the chamber, in this situation, not only soil respiration, but also plants respiration is also measured in your methods. this should be clearly stated in the introduction and methods part.

line 72: the authors state that percolation is related closely with soil, so drainage event may vary under different soil conditions. therefore, I thought you were studying percolation and drainage, well in the coming paragraph, for example in line100, you said you were studying irrigation method. this may be an error, there were difference between percolation, drainage and irrigation, they were defined differently. so you need to revise your saying in the introduction part. I think you could summarized these 3 definations as water management in paddy fields. in this way , things becomes better to understand.

line 103 such as ....etc, this saying is not allowed in a scientific paper, list all the parameters you measured here.

materials and methods:

line 127: you should expending the contents on measuring these soil characters. this paragraph is important, however, its too short now.

line 168: from the soil. this is not correct. CH4 and CO2 are not only emitted from soil but also from plants in your experiment.

line 174: I seriously doubt about this method, What is the accuracy of these two sensors? is the sensor accuracy enough form measuring especially when the fluxes were lower during the drainage period. pleasure provide more information about these 2 sensors, maybe a physical image is good to improve the feasibility.

line 215 the unit of GHGI is wrong i think. is it t CO2-eq ton-1?

Results and discussion:

1.Some notes should remove to the methods part, for example: line 261-265.

2. you dont have to provide too many Quantitative Numbers, instead, more comparasions should be presented. 

3. since you combined the discussion part into results and discussion. I think you use many references to discuss your results, however failed to compare your results with other papers published. many researchers in Japan, China and southeast countries has done the similar research. for example: DOI:10.1016/j.agwat.2023.108163  compared CH4 emission from paddy fields under different irrigation methods. it also provided clues why water saving irrigation decreased CH4 emission by  analyzing methanogenic bacteria and methane oxidizing bacteria.

Conclusion:

this part is fine, but a little long for a conclusion.

Reviewer 2 Report

Comments and Suggestions for Authors

Please see the attached reference

Comments on the Quality of English Language

It is appropriate. 

Round 2

Reviewer 1 Report

Comments and Suggestions for Authors

the authors` response has solved most of my concern, its ok to accept the manuscript.